# Induced Pluripotent Stem Cell-Derived Fibroblasts Efficiently Engage Senescence Pathways but Show Increased Sensitivity to Stress Inducers

**DOI:** 10.3390/cells13100849

**Published:** 2024-05-16

**Authors:** Marie-Lyn Goyer, Cynthia Desaulniers-Langevin, Anthony Sonn, Georgio Mansour Nehmo, Véronique Lisi, Basma Benabdallah, Noël J.-M. Raynal, Christian Beauséjour

**Affiliations:** 1Centre de Recherche du CHU Sainte-Justine, 3175 Côte Sainte-Catherine, Montréal, QC H3T 1C5, Canada; marie-lyn.goyer@umontreal.ca (M.-L.G.); cynthia.desau@gmail.com (C.D.-L.); anthony.sonn@umontreal.ca (A.S.); georgio.mansour.nehmo@umontreal.ca (G.M.N.); veronique.lisi.hsj@ssss.gouv.qc.ca (V.L.); basma.benabdallah.hsj@ssss.gouv.qc.ca (B.B.); noel.raynal@umontreal.ca (N.J.-M.R.); 2Département de Pharmacologie et Physiologie, Université de Montréal, Montréal, QC H3T 1J4, Canada

**Keywords:** iPSC-derived fibroblast, transformation, senescence, SASP, DNA repair

## Abstract

The risk of aberrant growth of induced pluripotent stem cell (iPSC)-derived cells in response to DNA damage is a potential concern as the tumor suppressor genes TP53 and CDKN2A are transiently inactivated during reprogramming. Herein, we evaluate the integrity of cellular senescence pathways and DNA double-strand break (DSB) repair in Sendai virus reprogrammed iPSC-derived human fibroblasts (i-HF) compared to their parental skin fibroblasts (HF). Using transcriptomics analysis and a variety of functional assays, we show that the capacity of i-HF to enter senescence and repair DSB is not compromised after damage induced by ionizing radiation (IR) or the overexpression of H-RAS^V12^. Still, i-HF lines are transcriptionally different from their parental lines, showing enhanced metabolic activity and higher expression of p53-related effector genes. As a result, i-HF lines generally exhibit increased sensitivity to various stresses, have an elevated senescence-associated secretory phenotype (SASP), and cannot be immortalized unless p53 expression is knocked down. In conclusion, while our results suggest that i-HF are not at a greater risk of transformation, their overall hyperactivation of senescence pathways may impede their function as a cell therapy product.

## 1. Introduction

To ensure the safety of iPSC-derived cells for clinical applications, it is essential to carefully study their genomic integrity and tumorigenic potential. Indeed, it has been observed that embryonic and iPSC lines can carry dominant negative mutations in cancer-associated genes [1,2]. Whether such mutations were originally present in parental cells or if they arose during reprogramming and were selected in vitro because they conferred a growth advantage is still unclear as contradictory results have been reported [3,4]. Yet, no study has so far evaluated the cell cycle checkpoint integrity and transformation potential of iPSC-derived cells when compared to their parental counterparts. Given the increasing interest in iPSC-based therapies, we believe this question is of utmost importance to ensure the long-term safety of clinical cellular products.

Independently of the starting cell type or the age of the donor, reprogramming efficiency is very low overall with less than 1% efficacy [5]. This may be explained by the observation that reprogramming leads to the activation of key tumor suppressor genes and that the expression of these genes must be downregulated to allow for the full reprogramming process [6,7,8]. For example, the activity of the p53–p21 pathway has been shown to be a barrier to iPSC generation. Therefore, its transient inactivation occurs during reprogramming at the expense of genomic stability [7,9,10]. Similarly, it has been shown that the Ink4a/Arf locus (encoding the p16 and p14/Arf tumor suppressor genes) acquires epigenetic marks and is completely silenced during iPSC reprogramming [8]. Whether the Ink4a/Arf locus and/or the p53–p21 pathways are fully reactivated in iPSC differentiated cells has never been thoroughly evaluated. Failure to completely reactivate these pathways would place cells at risk of transformation by compromising their ability to undergo senescence. Like apoptosis, cellular senescence is a phenotype that prevents damaged cells from proliferating [11,12,13]. p21 and p16, two tumor suppressors and cell cycle inhibitors, are arguably the best markers and inducers of senescence [14,15]. For example, p21 prevents cell cycle progression in response to DNA damage by a mechanism that entails an ATM-p53–p21 cascade [16,17,18,19]. In contrast, p16 often responds to DNA damage in a delayed manner [20,21]. Hence, defects in the functions of p53–p21 or p16 would lead to impaired senescence and compromise the safety of iPSC-derived cell therapies.

Based on this information, we wanted to investigate whether these pathways are fully functional in iPSC-derived cells. To achieve this, we used human fibroblasts (HF) isolated from the skin biopsies of three different donors and measured their ability to enter senescence and repair double-strand DNA damage compared to their autologous iPSC-derived counterparts (i-HF). These cellular functions were compared in response to the overexpression of the H-RAS^V12^ oncogene or following exposure to ionizing radiation (IR), two well-characterized inducers of senescence and DNA damage [21,22,23]. Our results showed that parental HF and i-HF clones display differences in their transcriptional profiles and secretomes upon the induction of senescence. Using a variety of cell cycle regulation and DNA repair functional assays, we observed that the ability of i-HF clones to enter senescence and repair DSB was not compromised, unlike their parental HF. Conversely, we observed a heightened sensitivity of i-HF clones to IR and oxidative/replicative stresses as shown by their inability to immortalize following their transduction with human telomerase reverse transcriptase (hTERT). This phenotype was donor dependent and majorly associated with the hyperactivation of the p53–p21 pathway and could be reversed by knocking down p53 expression. Our results suggest that i-HF do not exhibit an apparent increase in transformation risks. However, their overall increased sensitivity to stress should be considered when used as a cell therapy product.

## 2. Materials and Methods

### 2.1. Cell Culture, Reprogramming, Differentiation, and Immortalization

In accordance with the ethical committee of the CHU Ste-Justine (protocol 2017-1476), HF were obtained from skin biopsies collected from three different donors (HF1: adult male (41 years old), HF2: adult female (39 years old), and HF3: (fetal). All HF lines were derived in-house after dissociation using collagenase (Roche, Mannheim, Germany, 11088866001). HF were passaged weekly, and the medium was changed every 3–4 days in DMEM (Wisent, Saint-Jean-Baptiste, QC, Canada) supplemented with 10% fetal bovine serum (FBS) and 1% penicillin/streptomycin. The first passage was initiated at a population doubling (PD) of approximately 3, and stocks were frozen at low passages. Reprogramming was achieved through transduction with the integration-free Sendai virus-based Cytotune reprogramming kit (Life Technologies, Carlsbad, CA, USA, A16517) containing the 4 Yamanaka factors (Oct3/4, Sox2, c-Myc, Klf4). Cells were cultured in feeder-free conditions on Geltrex (Gibco, Grand Island, NY, USA, A14132-02) in Essential8 Flex+ medium (Life Technologies, A2858501) supplemented with Primocin (InvivoGen, San Diego, CA, USA, ant-pm-05) for two weeks until iPSC clones emerged. Single clones were manually picked and expanded on Matrigel (Corning, Bedford, MA, USA, 354230) in E8Flex for 4–6 passages on average until stable with no/minimum spontaneous differentiation. Colonies were screened using an EVOS™ XL Core Imaging System and spontaneous differentiation was eliminated via manual scraping before passaging. Isolated iPSC clones were characterized for pluripotency marker expression (SSEA4, Tra1-60, Sox2, and OCT4) via immunofluorescence using the Pluripotent Stem Cell 4-Marker Immunocytochemistry Kit (Life Technologies, A24881). iPSC-selected clones were kept in culture for a minimum of 8 weeks before starting differentiation to ensure stable pluripotency. Karyotypes were produced via G-banding and confirmed normal upon analysis by the CHU Ste-Justine cytogenetic department. hiPSC generation and characterization were performed in the iPSC—cell reprogramming core facility of CHU Sainte-Justine.

The differentiation process was initiated by the formation of embryoid bodies (EB) in APEL2 medium (StemCell, Vancouver, BC, Canada, 05275) supplemented with 10 ng/mL of FGF2 (Peprotech, Cranbury, NJ, USA, 100-18B) and 5 ng/mL of TGFβ (Peprotech, 100-21). On day 7, EB were transferred onto Geltrex-coated plates and cultured in DMEM +10% FBS +1% penicillin/streptomycin, and cytokines were supplemented for eight additional days. The cells were cultured similarly to the HF, apart from being plated on 1% Geltrex-coated plates to prevent cell death during the early expansion phase. Throughout all culture steps, cells were maintained at a normoxic (5%) oxygen level. To immortalize the cells, transduction was initially carried out overnight using lentiviruses expressing hTERT followed by transduction with lentiviruses expressing either shRNAp53 or CDK4 m in the subsequent passages. The shRNAp53 lentivector was a gift from Dr. Francis Rodier (CHUM, Montréal, QC, Canada) and the CDK4 m lentivector was described previously [15].

### 2.2. Immunofluorescence

Cells were plated at 50–70% confluence on a coverslip coated with Geltrex 1% and left in culture for at least two days. The cells were fixed for 10 min with 4% paraformaldehyde (PFA), permeabilized with PBS 0.2% Triton X-100 for 15 min, and blocked with PBS 10% FBS for 30 min with three washes in between steps. For hTERT immunofluorescence, cells were blocked and permeabilized simultaneously with PBS containing Triton 0.1% and BSA 1% for 1 h at RT. SSEA4, TRA1-60, Vimentin, FSP1, or hTERT primary antibodies (See Appendix A) were incubated overnight at 4 °C. Cells were incubated with Alexa 594 (Invitrogen, Eugene, OR, USA) secondary antibody for 30 min, washed with DAPI (0.5 µg/mL), and then mounted on a slide with Vectashield before pictures were taken with an Olympus BX51 microscope using a Qimaging Retiga 2000 R camera (Burnaby, BC, Canada). Brightfield images were taken using the EVOS M5000 Imaging System.

For the staining of bromodeoxyuridine (BrdU, Sigma, Darmstadt, Germany, B5002), cells were plated at 40–50% confluence on a coverslip coated with Geltrex 1% and irradiated with a dose of 2.5 or 5 Gy. Two days later, BrdU was added to the culture medium at a concentration of 10μM. Cells were fixed on day 6 with PFA 4%, and then DNA was denatured with HCl 1N for 30 min at 37 °C and neutralized with a borate buffer of 0.1 M pH 8.5 for 10 min at room temperature (RT). Following washes with PBS, cells were incubated with an anti-BrdU antibody (Appendix A) diluted in PBS 1% BSA overnight at 4 °C. Secondary antibody mouse Alexa 488 (Invitrogen) was added following washes at 1:750 in PBS containing 1% BSA for 1 h at RT. Cells were washed, stained with DAPI, and mounted on a slide with Vectashield before pictures were taken. Counts were performed using ImageJ (v2.15.0) on 3 different areas of the coverslip covering an average of 150 cells each.

For DNA damage detection by γH2 AX and 53BP1 foci, cells were seeded on Geltrex 1% coated coverslips and irradiated at 1 Gy two days later. Cells were fixed with PFA 4% 1 h, 8 h, 24 h, and 48 h post irradiation and permeabilized for 10 min with PBS containing 0.2% Triton. Blocking was performed for 1 h with PBS containing 5% goat serum and 1% BSA. γH2AX and 53BP1 primary antibodies (see Appendix A) were added and incubated overnight at 4 °C. Alexa 488/594 secondary antibodies (Invitrogen) were incubated 1:750 for 1 h, washed with DAPI, and mounted on slides before pictures were taken. Counts were performed using ImageJ on 3 different zones of the coverslip covering an average of 150 cells each.

### 2.3. Senescence Induction and Detection

Senescence was induced either by transduction of an H-RAS^V12^ lentiviral vector or by exposure to 15 Gy ionizing irradiation (Faxitron X-Ray CP-160). The transduction control (Mock) contained a puromycin selection gene only. Senescence was assessed 10 days later by evaluating cell morphology and the absence of proliferation, senescence-associated beta-galactosidase (SA-β-gal), and by measuring the SASP (see Section 2.5 below). SA-β-gal expression was confirmed using the previously described protocol by Itahana et al. [24]. Pictures were taken using the EVOS M5000 Imaging System, and cell counts were performed using ImageJ on five different pictures taken from randomly selected areas within the well.

### 2.4. Colony Formation Assay (CFA)

After senescence was induced, cells were plated at a density of 104 cells per 100 mm dish, and the medium was changed every 3–4 days over three weeks. Cells were then washed with PBS, fixed for 10 min with PFA 4%, and stained with a solution of 0.5% (*v*/*v*) crystal violet in MeOH for an hour before being rinsed at room temperature with water to remove excess dye. The total number of colonies was counted manually by sight.

### 2.5. Conditioned Media Collection

On day 9 after senescence induction, the time at which the SASP was shown to be fully deployed, HF were washed 3 times with PBS, and DMEM without FBS was added to cells at 85–90% confluency. Conditioned media were collected 24 h later and filtered on a 0.22 μM filter syringe before being flash-frozen on dry ice. Cells were counted at the time of media collection to later normalize the concentration of secreted factors. Samples were sent to and analyzed by Eve Technologies Corporation (Calgary, AB, Canada) using the Human Cytokine/Chemokine Panel A 48-Plex Discovery Assay®.

### 2.6. qPCR

Cells were washed three times with PBS and then lysed directly on 100 mm culture dishes on ice using RLT buffer (RNeasy kit, Qiagen, Germantown, MD, USA, 74106) + 1% β-Mercaptoethanol (Sigma, M3148) and detached with a cell scraper. Cells were collected and homogenized via vortexing before extracting RNA following the RNeasy kit protocol from Qiagen. Total RNA concentration and purity were assessed using a NanoDrop ND-1000 spectrophotometer (Wilmington, DE, USA). Reverse transcription was performed using the QuantiTect Reverse Transcription kit (Qiagen, 205314). Forward and reverse primers (500 nM) were loaded in a 96-well qPCR plate (Roche, 04729692001) along with 10 ng of cDNA and 1× Power SYBR™ Green PCR Master Mix (Applied Biosystems, Vilnius, Lithuania, A25742). The samples were loaded in duplicate and normalized to the 18S reference gene. The plate was read using a LightCycler^®^ 96 instrument (Roche).

### 2.7. RNA Sequencing

RNA was extracted from cultured HF or i-HF using the RNeasy kit (Qiagen). RNA quality control, library preparation, and sequencing were performed using the Génome Québec platform. RNA quality was assessed using the Perkin Elmer LabChip (RIN > 6.5). Illumina Stranded mRNA Prep was used for library preparation. Paired-end sequencing (PE100) was performed on the Illumina NovaSeq platform, achieving an average sequencing depth of 25 million reads per sample. Quality control was performed using FastQC. Reads were trimmed using trimmomatic (v0.39), aligned to the human genome GRCh38 using STAR2.7.9, and quantified using RSEM 1.3.1. Differential expression analysis and Student’s *t*-test statistical analysis were conducted using R (v4.3.1) and RStudio (v2023.09.01+494) software with the ggplot2 package (v3.4.4) for data visualization. The gene set enrichment analysis (GSEA) results were analyzed with the fgsea package (v3.18) from Bioconductor using reactome_cell_cycle, reactome_DNA_repair, and reactome_regulation_of_TP53_activity gene sets from the Molecular Signatures Database (MsigDB, v2023.2.Hs). GO_Biological_Process_2023 from Enrichr packages (v3.2) was used for gene ontology analysis. The raw sequencing data generated in this study have been deposited in the Gene Expression Omnibus (GEO) database under accession number GSE266663.

### 2.8. Western Blot

Cell pellets were washed and lysed in a homemade radioimmunoprecipitation assay (RIPA) buffer with 1% protease inhibitor (Sigma, P8340) for total protein extraction. Proteins were quantified using the Rapid Gold BCA Protein Assay Kit (Thermo Fisher, Rockford, IL, USA, A53226), and 20–30 μg of protein from each sample were loaded on a gel. Gel electrophoresis was conducted at 110 V for 1 h using 4–15% Mini-PROTEAN^®^ TGX™ Precast Protein Gels (Bio-Rad, Hercules, CA, USA, 4568084), followed by wet transfer using a PVDF membrane (Merck Millipore, Cork, Ireland, IPVH00010) at 65 V for 2 h at 4 °C. Membranes were blocked in a tris-buffered saline, 0.1% Tween (TBST) solution containing 3% BSA, and immunoblotted overnight at 4 °C (see Appendix A for antibodies). Horseradish peroxidase (HRP)-coupled secondary antibodies (Santa Cruz, Dallas, TX, USA) were incubated for 1–1.5 h with the membranes, and protein detection was performed on G:BOX Chemi XRQ using Clarity Western ECL Substrate (Bio-Rad, 170-5060).

### 2.9. DNA Repair Assay

HF containing a single chromosomally integrated reporter construct to analyze DSB repair were generously offered by the laboratory of Pr. V Gorbunova. The sequence and integration of the non-homologous end joining (NHEJ) reporter plasmid, as well as the protocol for I-SceI nucleofection and DNA repair analysis by FACS, are described in the work of Seluanov et al. [25]. The HF were first reprogrammed and then differentiated using the same techniques described above. The day following nucleofection, a subset of parental (HF4) and iPSC-derived (i-HF4) cells were treated with 10 μM of suberoylanilide hydroxamic acid (SAHA) for 24 h. On day 4 post nucleofection, the expression of DsRed (transfection control) and GFP (NHEJ repaired cells) was assessed using FACS BD Fortessa, and data analysis was performed using Flowjo software (v10.8.1). A ratio was calculated using the percentage of GFP+ on DsRed+ cell expression to evaluate the level of DNA repair by NHEJ. DsRed+ cells were also sorted using FACS and lysed for DNA extraction followed by PCR amplification of the region encoding the adenoviral exon flanked with the I-Sce1 cut sites (see Appendix A for the sequence). PCR products were loaded on 2% agarose gel. Repaired (429 bp) and unrepaired (635 bp) bands were detected with RedSafe dye (INtRON, 21141) on G:BOX Chemi XRQ. Quantification of DNA band signals was performed using ImageJ and the percentage of repair was calculated as the signal from the repaired band over the sum of the repaired and unrepaired band signals.

### 2.10. Oxidative Stress Assay

Cells were plated in a 96-well plate (5 × 10^3^/well) and were treated 2 days later either with H_2_O_2_ (2.5 μM or 5 μM) or menadione (100 μM or 200 μM) for 1 h, and cell toxicity was assessed after 1 h, 4 h, and 8 h. At each timepoint, the supernatant and trypsinized cells were collected and transferred into a 96 V-well plate for staining with 3 μM propidium iodide (Sigma, P4864). Cell death was detected using FACS BD Fortessa, and data analysis was performed using Flowjo software.

### 2.11. Statistics

Graphic design and statistical analyses were conducted using GraphPad Prism version 9.3.1 unless otherwise stated. The significance level was set at α = 0.05.

## 3. Results

### 3.1. i-HF Are Phenotypically Similar to Their Parental HF but Display a Distinct Transcriptomic Profile

To determine the integrity of senescence pathways in i-HF, we sought to compare i-HF lines with their parental autologous counterpart. To achieve this, we collected skin HF from three donors (two adults and one fetal donor) and reprogrammed them into iPSC lines (hereafter referred to as i-HF1, i-HF2, and i-HF3. respectively) using non-integrating Sendai viruses. Two iPSC clonal lines for each donor were induced into forming embryonic bodies and subsequently differentiated back into HF using a defined protocol containing FGF2 and TGFβ (Figure 1a). All i-HF lines showed a similar morphology and expressed several markers shared by their parental counterparts (Figure 1b–d). However, i-HF were smaller in size overall and had a less elongated morphology, particularly at low PD. All cell lines, except for the i-HF1 clone 1, were able to proliferate between 30–40 PD before entering senescence (Figure 1e). Moreover, i-HF clones generally had a shorter doubling time compared to their parental HF (Figure 1e). The transcriptomic analysis revealed that i-HF are highly distinct from their parental HF and show more heterogeneity among themselves (Figure 1f). The gene ontology analysis revealed increased mitochondrial activity, ribosomal biogenesis, and translation in i-HF, which is consistent with their overall shorter doubling time (Appendix A). Also distinctive of i-HF is their decreased capacity to produce an extracellular matrix (Appendix A), hence our use of Geltrex-coated plates to facilitate their expansion (see the Materials and Methods section). Gene set enrichment analysis also showed increased cell cycle activity in i-HF lines and a tendency for increased TP53 activity with a marked difference observed for donor 2 (Appendix A).

We next wanted to characterize the molecular pathways involved in inducing senescence. To achieve this, we collected proteins from all cell lines at early passage (range 7–13 PD) or at a passage close to senescence (range 19–43 PD depending on the clone) and measured the expression of p16 and p21 using Western blots. These two proteins are known to be involved in stress-induced and replicative senescence, respectively [15]. Our results showed that the HF1 line is more prone to upregulate the expression of p16 than lines from the other donors, even at early PD (Figure 1g). This relatively high p16 expression pattern holds for i-HF1 lines, particularly clone 1 which senesce at around 20 PD (Figure 1g). Alternatively, cell lines from donors 2 and 3 tended to rely more on p21 or other mechanisms with little contribution from p16, except for i-HF3 clone 2 (Figure 1g). Overall, it seems that the choice to undergo replicative or stress-induced senescence is donor dependent and that this trait is conserved in i-HF lines. Together, these results show that it is possible to successfully differentiate iPSC into i-HF that are phenotypically similar to their parental counterparts and that i-HF lines show on average only a modest increase in their proliferative potential despite cycling faster. However, our transcriptional analysis revealed broad differences in gene expression, including overexpressed cell cycle genes in i-HF, which further supports the need to better characterize these cells.

### 3.2. p53-Related Pathways Are Activated in a Diverse and Generally Heightened Manner in i-HF Cell Lines

p53 activation serves as a critical checkpoint to ensure that damaged cells are either repaired or eliminated. Given its pivotal role in key cellular functions and its downregulation during reprogramming, we wanted to investigate the activation of its related pathways in i-HF lines. Transcriptional data indicated that, in response to IR, there are no significant changes in the expression of genes regulating p53 activity in i-HF compared to their parental HF (Figure 2a). Within the list of differentially upregulated genes (DEGs) induced by IR and known to be regulated by TP53, 10 out of 26 were shared between i-HF and HF, including MDM2 and CDKN1A (p21), as shown in a two-set Venn diagram (Figure 2b,d and Appendix A). Still, HF exhibited 15 unique DEGs compared to only one in i-HF (Figure 2b,d and Appendix A). This discrepancy may be partially explained by the heterogeneity in gene expression observed between i-HF and parental HF cell lines in response to IR as represented by principal component analysis (PCA) (Figure 2c). The analysis also showed that following DNA damage, HF lines still cluster more closely compared to i-HF lines (Figure 2c). The latter is also much further spread on the PC2 axis, particularly for donor 3, where the irradiated sample is notably distant from its respective control (Figure 2c). We confirmed these transcriptional data by measuring the relative gene expression of GADD45A and CDKN1A at various timepoints after exposure to IR using qPCR (Figure 2d,e). Finally, we also investigated the expression of p21 and p53 proteins, as well as the activated form via phosphorylation at serine 15 (p-p53(Ser15)) in response to IR. Phosphorylation on serine 15 mediated by ataxia-telangiectasia mutated and Rad3-related kinases is the primary target of the DNA damage response on p53, stabilizing it and preventing its degradation [26]. As expected, overall expression levels of p21 were increased in all cell lines in response to IR although at variable levels (Figure 2f,g and Appendix A). Similarly, the level of p-p53(Ser15) normalized to total p53 was sharply induced by IR but not significantly different between the HF and i-HF lines (Figure 2f,g and Appendix A). However, we observed increased expression of p21 and p53 at basal levels in i-HF compared to the parental cells (Figure 2f,g and Appendix A). Taken together, our results suggest that i-HF lines are heterogeneous for the activation of pathways involving p53 functions, yet the principal p53 effectors do not appear to be compromised and are sometimes even increased in response to IR.

### 3.3. i-HF Efficiently Enter Senescence Following DNA Damage

We next wanted to investigate whether i-HF were as proficient as their parental HF in halting proliferation and undergoing senescence in response to DNA damage. To achieve this, we first measured the proportion of cells that ceased incorporating BrdU after being exposed to IR (2.5 or 5 Gy), a known inducer of DNA DSB. While all cell lines reduced BrdU incorporation in response to IR, we observed that i-HF from donors 1 and 3 were more inhibited (Figure 3a). We then evaluated if i-HF lines could permanently cease proliferation and enter senescence in response to two validated inducers of senescence. One is generated by exposure to a high dose of IR and the second is induced by the expression of an oncogene (known as oncogene-induced senescence). Indeed, constitutive activation of an oncogene such as H-RAS^V12^ is known to produce a persistent DNA damage response (DDR) caused by uncontrolled mitotic signals and a loss of nuclear integrity [27]. Therefore, i-HF and their parental lines were exposed to either IR at the dose of 15 Gy or transduced with H-RAS^V12^ lentiviral particles. Proliferation was quantified using weekly cell counts and the ability to form colonies when plated at low cell density. All lines efficiently halted cell proliferation in response to both senescence inducers with no significant difference between the parental and i-HF lines (Figure 3b–d). Further confirmation that cells have entered senescence was obtained as a strong proportion showed an increased expression of SA-β-gal, a lysosomal enzyme commonly used as a marker of senescence (Figure 3e,f). Of note, i-HF from donor 1 showed a reduced amount of clone outgrowth and higher SA-β-gal activity at basal level or after the introduction of an empty viral vector (Mock), which is consequent to their reduced growth potential (Figure 1e). Moreover, in line with our BrdU incorporation data, a slightly lower dose of IR (12 Gy) was sufficient to completely arrest clonal outgrowth in i-HF lines but not in parental lines (Appendix A). Overall, these results not only suggest that i-HF lines are fully capable of activating the senescence phenotype but that these lines are perhaps even more sensitive to DNA damage.

To investigate potential variations in the level of stress triggered by these senescence inducers, we looked at the senescence-associated secretory phenotype (SASP), another hallmark of senescent cells. The SASP is a complex phenotype mediated by key signaling pathways such as NF-κB, and its intensity is kept in check in part through the action of p53 [28,29]. The SASP profile is known to vary depending on the inducer and can have important physiological consequences by inducing inflammation and attracting immune cells among many things [28,30]. We thus measured the SASP in the conditioned media of cells 10 days following the induction of senescence (IR and H-RAS^V12^ induced) using multi-plex cytokine arrays and by normalizing the secretion to the total number of cells. Our results demonstrated a more pronounced overall and diverse SASP profile in H-RAS^V12^ compared to IR-induced senescent cell lines (Figure 4). However, in i-HF lines, the SASP was much lower overall in response to H-RAS^V12^-induced senescence, particularly for the secretion of GM-CSF, MIP-1b, MCP-3, CXCL9, PDGF, EGF, fractalkine, and TNFα (Figure 4b). The reason for this is unknown as the secretion of the same cytokines (i.e., EGF and fractalkine) and some others (i.e., FGF-2, IL12p40, MCP-1, and INF-γ) was found to be increased in i-HF lines in response to IR (Figure 4b). Hence, it appears that i-HF lines are more sensitive overall to IR compared to their parental lines. Collectively, this suggests that while i-HF can efficiently activate the senescence phenotype, they seem to respond differently from their parental cell lines in response to a specific inducer of senescence.

### 3.4. DNA Double-Strand Break Repair Is Not Compromised in i-HF Lines

Sustained irreparable DSB foci have been shown to promote and maintain cellular senescence and its SASP [31,32]. Given that i-HF have been shown so far to be more sensitive to IR-induced DNA damage, we sought to determine the efficacy of DSB repair in i-HF. We first analyzed the impact of IR on the upregulation of genes associated with DNA repair between i-HF cell lines and their parental counterparts. Transcriptional analysis revealed no statistical differences in the expression of DNA repair-related genes (*p* = 0.957) 8 h following DNA damage (Figure 5a). To directly measure the kinetics of DSB repair, we irradiated our cell lines at a dose of 1 Gy to induce low and thus quantifiable numbers of DSB. To achieve this, we counted the number of 53BP1/γH2AX foci per nuclei, two proteins involved in the initiation of the repair process, at different timepoints covering 48 h post exposure to IR (Figure 5b,c). As expected, the number of 53BP1/γH2Ax foci gradually declined over time in both HF and i-HF for each of the three donors (Figure 5c). No significant differences were observed, except that i-HF lines (particularly donor 1 and 2) appear to have overall fewer DNA damage foci in the absence of IR, and i-HF from donor 1 could repair its foci faster than its parental counterpart (Figure 5c). Hence, transcriptomic and functional data suggest that i-HF are fully functional at repairing DSB in response to IR.

To confirm these results, we also took advantage of a GFP reporter DNA system to measure the rate of DNA repair in the form of non-homologous end-joining (NHEJ) [24]. NHEJ represents one of the two primary DNA repair mechanisms of DSB along with homologous recombination (HR). NHEJ largely predominates over HR as the mechanism used by most cells in the G1 phase of the cell cycle, when a sister chromatid is not readily available [33]. To measure the efficiency of NHEJ in i-HF, we generated a new iPSC line from HF (named HF4) containing a single insertion of a GFP reporter construct. This construct is constituted by two I-SceI endonuclease DSB sites which restore GFP expression when repaired using NHEJ (see Figure 5d). Hence, both the parental HF4 and the i-HF4 were nucleofected with plasmids expressing a DsRed reporter gene or the I-SceI endonuclease. Of note, to avoid a clonal effect, we worked with a pool of three reprogrammed iPSC clones that were differentiated in bulk. Successful DNA repair was quantified using flow cytometry based on the ratio of GFP-expressing cells to the total number of cells expressing DsRed. This step is necessary to correct the efficacy of nucleofection. Unexpectedly, we observed almost no GFP expression in i-HF4 compared to when using the parental HF4 (Figure 5e). These results lead us to hypothesize that i-HF4 may have acquired epigenetic modifications during the reprogramming process that led to the inactivation of the CMV promoter responsible for GFP expression. Indeed, cells engineered to express a gene of interest often undergo silencing during reprogramming and differentiation [34]. To verify this hypothesis, we treated the cells with the histone deacetylase inhibitor suberoylanilide hydroxamic acid (SAHA). However, while the treatment did restore GFP expression, it failed to restore it to the level observed in parental HF4 (Figure 5e). To overcome this possible technical limitation, we instead isolated DNA from FACS-sorted DsRed nucleofected cells and amplified via PCR the DNA region covering the NHEJ sites (see Appendix A). Agarose gels revealed, as expected, an unrepaired band at 635 bp, and in cells receiving the I-SceI enzyme, a repaired band at 429 bp, the latter confirming deletion of the adenoviral exon and ligation of DNA by NHEJ (Figure 5f). Using this approach, we found that the proportion of cells that underwent NHEJ repair was similar in both cell lines to the ratio of the two DNA bands (429 bp/635 bp) was non-significantly different (Figure 5g). These results suggest that the NHEJ repair machinery is not impaired in the polyclonal i-HF4 cell line.

### 3.5. Inability to Immortalize i-HF Lines Is p53 Dependent

So far, we have shown that despite their heterogeneity, i-HF lines are fully functional at activating the senescence program and repairing DSB and have functional p53-related effector pathways. Still, we next wanted to evaluate if i-HF lines were more prone to transformation compared to their parental cell lines. This is an important question in the context where iPSC-derived cells would engraft long-term into the host. Hence, to test this hypothesis, we first transduced i-HF lines with a lentiviral vector expressing the hTERT gene and measured their immortalization potential. All cell lines were successfully transduced with, on average 40% of the cells expressing hTERT initially, a proportion that rapidly reached close to 100% with subsequent passages. Surprisingly, none of the i-HF lines could be immortalized with hTERT, as opposed to their parental HF (Figure 6a). Given that i-HF lines showed hyperactivated p53 and p21 effector functions at basal level (Figure 2), we thought that they may be more sensitive to the stress generated by prolonged cell culture conditions. Indeed, we found that i-HF lines from donors 1 and 3 were more prone to increased cell death following exposition to hydrogen peroxide (H_2_O_2_) and menadione, respectively (Appendix A). H_2_O_2_ acts as a precursor for other reactive oxygen species (ROS) while menadione generates ROS through redox cycling, especially impacting mitochondria through electron leakage [35]. Considering these results, we investigated if we could overcome this heightened sensitivity to stress by silencing the expression of p53 (using a shRNA against p53) or introducing a mutant, constitutively active form of CDK4 (CDK4m), which overcomes p16-mediated cell cycle inhibition (Figure 6b). We found that i-HF can be immortalized with hTERT only when knocking down p53 (Figure 6c). The overexpression of CDK4m had no beneficial effect on i-HF lines from donors 2 and 3 as they showed low p16 expression levels upon senescence (Figure 1g). Surprisingly, the overexpression of CDK4m was also insufficient to bypass the replicative senescence of i-HF1 cl2, which express higher levels of p16. Of note, i-HF1 clone 1 was not evaluated as this clone showed very limited growth potential (Figure 1e and Figure 6c). In conclusion, i-HF display an increased sensitivity to oxidative and/or replicative stress and can be immortalized only if p53 expression is knocked down.

## 4. Discussion

Despite the emergence of therapeutic applications using iPSC-derived cells, the integrity of their tumor suppressive pathways remains mostly unknown. So far, multiple groups have looked at the genetic and epigenetic instability, as well as DNA repair capacity, in iPSC versus somatic cell lines [36,37,38]. However, no previous studies had compared the integrity of iPSC-derived cells to their autologous primary cell line. Our study combined transcriptional and protein expression analyses with several functional assays using cell lines from three different donors to determine responses to oncogenic, IR, oxidative, and replicative stress. Fortunately, our results demonstrate that i-HF are not at a greater risk of transformation, at least when evaluated for their ability to engage senescence pathways and repair DSB in response to DNA damage. These results are important as they suggest that the reprogramming procedure, despite transiently selecting for cells with inactivated TP53 and CDKN2A tumor suppressor genes, does not subsequently interfere with the function of these genes. However, one needs to account for the fact that iPSC-derived cell functions likely depend on the differentiation protocol used, and the results we obtained may have been different had we used a different differentiation protocol or DNA-damaging insult. However, it is reasonable to believe that other types of drugs known to induce DNA damage and senescence (i.e., doxorubicin) would have led to similar conclusions.

Still, despite the apparent integrity of i-HF, all three lines exhibited a very distinct transcriptional profile compared to their parental lines. The lack of terminal differentiation and maturation into skin-derived fibroblasts may explain such differences. Indeed, gene ontology (GO) analysis revealed the upregulation of mitochondrial activity coupled with increased translation and other GO terms, indicating a high protein turnover rate in i-HF lines. Extensive metabolic reconfiguration, essentially through mitophagy, is a crucial phenomenon reported to occur during the transition of cell fate. This process arises as a result of transitioning from glycolysis into an oxidative phenotype, driven by the increased energy demand associated with cell differentiation [39,40,41,42] consistent with not only the elevated proliferation rate observed in most i-HF cell lines but also increased sensitivity to IR and ROS. It is well known that highly proliferating cells are more affected by IR-induced DNA damage. This may partially explain why i-HF, characterized by heightened mitochondrial activity, appears more stressed with elevated expression of p53 effector genes observed at the basal level. This hypothesis is supported by our results showing that i-HF lines could not be immortalized following transduction with hTERT unless the expression of p53 was knocked down. While this may be beneficial overall for preventing cell transformation, such an increase in p53 activity and sensitivity to stress may have downstream deleterious consequences. Indeed, we observed that i-HF lines show overall heightened expression of SASP factors in response to IR, particularly MCP-1, which plays a role in monocyte chemotaxis and oxidative stress. The SASP is known to be composed of a mixture of proinflammatory molecules that can lead to the onset of age-related degenerative diseases [43]. A different secretory phenotype could have physiological implications, such as aberrant immune reactions, following the injection of iPSC-derived cells in vivo. For example, many diseases affecting the brain, joints, and bones are associated with elevated levels of MCP-1 [44]. However, considering the relatively low number of iPSC-derived cells that would be transplanted, we speculate that this should not have important physiological consequences, particularly given that the SASP of i-HF at basal levels (in the absence of DNA damage) was similar to the SASP of the HF lines. Moreover, it is unknown at the moment as to why i-HF lines displayed an attenuated SASP profile compared to HF lines in response to H-RAS^V12^-induced senescence. Again, the overall increased metabolic activity and doubling time of i-HF lines may make them less sensitive to hyperproliferative signals induced by H-RAS^V12^.

We restricted our investigation to fibroblasts as a model for comparing parental and iPSC-derived cells due to their ease of acquisition, culture, and reprogramming. It is important to note that the iPSC and i-HF generated in this study were employed as proof of concept and were not designed to meet clinical-grade standards. Whether other cell types, for example, iPSC-derived epithelial cells, would show the same ability to engage senescence pathways and repair DSB remains to be determined. To answer this question, one would need to successfully derive parental cells that can be expanded in vitro and used as controls for iPSC-differentiated cells. For example, we foresee that this could be achieved using endothelial cells, keratinocytes, mesenchymal stem/stromal cells, or myoblast progenitor cells. However, measuring senescence integrity pathways may not be necessary for cells that do not proliferate upon differentiation such as cardiomyocytes, neurons, or hepatocytes. Indeed, we previously showed that only iPSC-derived hepatic progenitor cells, but not hepatocytes, can be transformed in vitro [45]. Of note, we did not measure the propensity of i-HF lines to form tumors in response to exposure to carcinogens. This is because we concluded that it was not worthwhile to pursue this approach due to the inability to immortalize i-HF lines in vitro following the introduction of the hTERT gene.

## 5. Conclusions

In conclusion, considering the diversity of differentiation protocols and cell sources available, we argue that a more comprehensive characterization of iPSC-derived cells, both in vitro and perhaps in vivo, is necessary as therapies utilizing iPSC-derived products are becoming increasingly prevalent. Moreover, given the great disparity in the transcriptional profiles we observed between i-HF and their parental HF, it may be beneficial to evaluate the functionality of key tumor suppressor genes following various types of DNA damage as a routine test before moving to the clinic.

## Figures and Tables

**Figure 1 cells-13-00849-f001:**
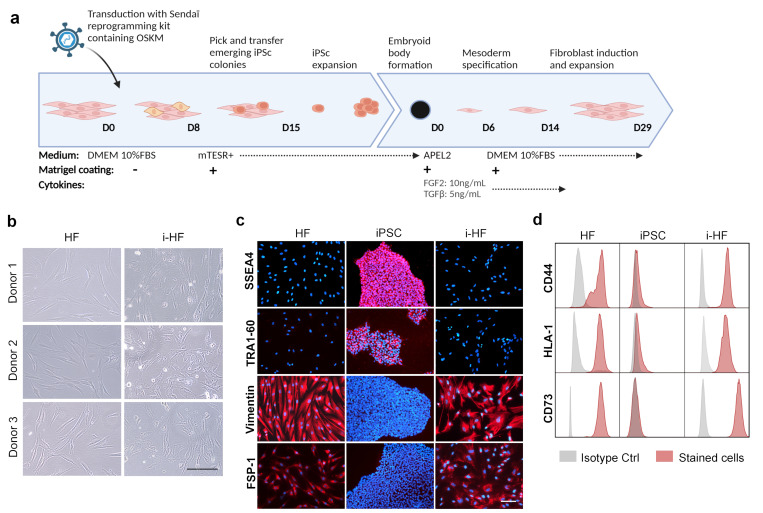
Phenotypic and transcriptomic characterization of i-HF and their parental HF. (**a**) Timeline representing reprogramming and differentiation protocols of HF obtained from skin biopsies. (**b**) Representative brightfield images showing HF and i-HF morphology taken from 3 different donors. Scale = 100 μm. (**c**) Representative images showing the expression of iPSC (SSEA4 and TRA1–60 in red) and HF (Vimentin and FSP-1 in red) markers via immunofluorescence on HF and i-HF from donor 1 are shown. DNA was stained with DAPI (in blue). Scale = 100 μm. (**d**) Expression of markers (CD44, CD73, HLA-1) typically found on HF as evaluated via flow cytometry on HF and i-HF from donor 1. (**e**) Growth curves representing cumulative population doubling (PD) over time of HF and i-HF derived from 2 different iPSC clones (cl). (**f**) Principal component analysis (PCA) illustrating the clustering patterns of individual samples based on gene expression profiles derived from RNA-seq normalized read counts. Sample-to-sample distances between donors (D1–D3) are illustrated for HF and i-HF (cl1 for each donor was used). (**g**) Protein expression of p16 and p21 at an early or late PD of the indicated cell lines. Late PD (ranging from 19 to 43 PD) was established when the cells stopped proliferating for 3 consecutive passages.

**Figure 2 cells-13-00849-f002:**
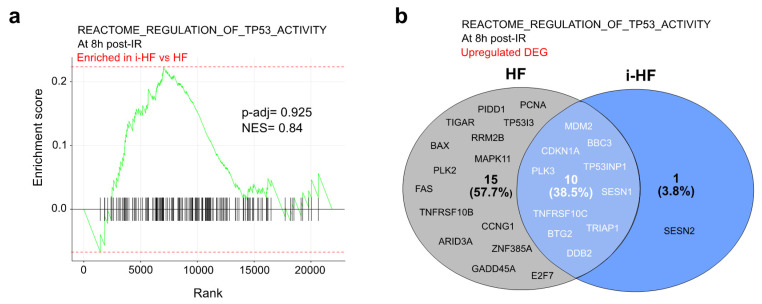
i-HF activate p53 effector genes following IR-induced DNA damage but show heterogeneous expression between donors. (**a**) Gene set enrichment analysis (GSEA) and enrichment of gene sets associated with the Reactome_Regulation_of_TP53_Activity was conducted using the pool of the three i-HF compared to HF donors at 8 h post exposure to 15 Gy IR. The adjusted *p*-value (P-adj) and normalized enrichment score (NES) are shown. (**b**) Venn diagram representing differentially expressed genes (DEGs) from the previously mentioned biological pathway that are upregulated after IR only in HF (grey), only in i-HF (blue), or both. (**c**) PCA illustrating the clustering patterns of individual samples based on gene expression profiles derived from total RNA-seq normalized read counts. Sample-to-sample distances between donors (D1–D3) are illustrated for HF and i-HF at basal level (Ctrl) and 8 h post IR. (**d**) Number of mRNA transcripts per million (TPM) mapped reads from RNA-seq data of each of the three donors of HF and i-HF at basal level (Ctrl) and 8 h post IR. Representative genes were selected from each section of the Venn diagram. The lines represent the mean value. Data were analyzed using multiple unpaired t-tests between HF and i-HF for each group. (**e**) Relative gene expression of GADD45A and CDKN1A by qPCR at different times (8, 24, and 48 h) post IR. Data were normalized on 18S gene expression, and fold change is relative to the HF Ctrl of the related donor. Data analysis was performed using two-way analysis of variance (ANOVA) followed by Fisher’s LSD post hoc test. Shown is the mean ± SEM of n = 3 different experiments. (**f**) Protein expression of p21, p53, and phosphorylated p53 at serine 15 (p-p53(Ser15)) at 8, 24, and 48 h following exposure to IR. Shown is a representative membrane for the expression from donor 1. Vinculin was used as a loading control. (**g**) Quantification of p21 and p53 protein expression normalized on vinculin signal and quantification of the ratio of p-p53(Ser15)/p53 protein expression from donor 1. Data analysis was performed using two-way ANOVA followed by Fisher’s LSD post hoc test. Shown is the mean ± SEM of n = 3 individual Western blots. ns: not significant, *: *p* < 0.05, **: *p* < 0.01, ****: *p* < 0.0001.

**Figure 3 cells-13-00849-f003:**
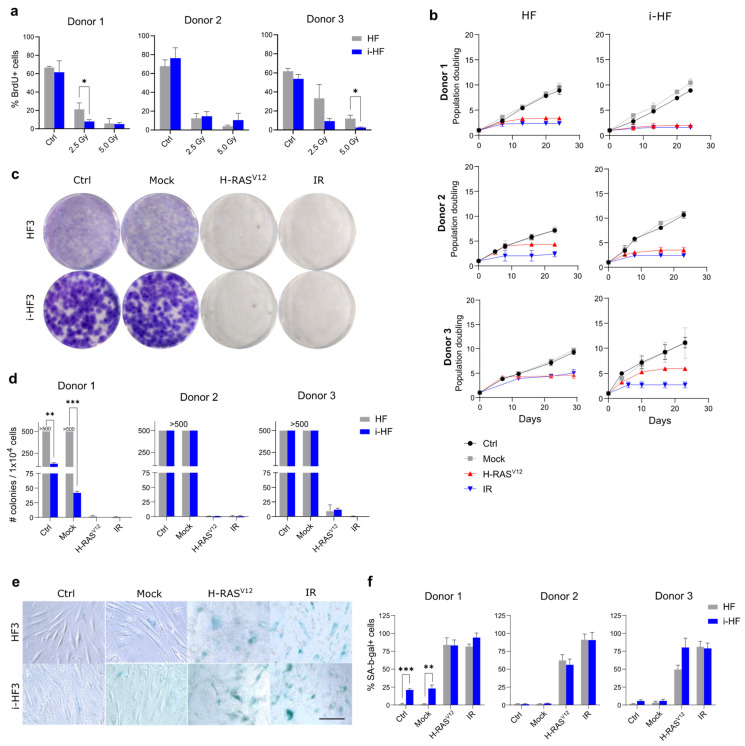
i-HF enter senescence as efficiently as their parental HF following DNA damage. (**a**) The proportion of HF that have incorporated bromodeoxyuridine (BrdU) after DNA damage induced by IR (2.5 and 5 Gy). Data analysis was performed using two-way ANOVA between HF and i-HF for each dose of IR followed by Fisher’s LSD post hoc test. Shown is the mean ± SD of n = 3 independent experiments. (**b**) Cell growth as measured by cumulative cell PD over a period of three weeks following DNA damage induced by the transduction of H-RAS^V12^ lentiviral particles or following exposure to 15 Gy IR (n = 3). (**c**) Representative images of a colony formation assay (CFA) performed three weeks after senescence was induced (as described in panel (**b**)) using HF3 and i-HF3 clone 1. (**d**) Number of colonies counted from the CFA described above for all 3 donors. A threshold value of 500 colonies was estimated when the cells were deemed too confluent to be counted. Shown is the mean ± SD of n = 3 independent experiments. (**e**) Representative images showing the expression of SA-β-gal ten days after senescence was induced on HF3 and i-HF3 clone 1. Scale = 100 μm. (**f**) Quantification of the proportion of HF and i-HF from all 3 donors expressing SA-β-gal after senescence induction. Shown is the mean ± SD of n = 3 independent experiments. Data from both CFA and SA-β-gal assays were analyzed using multiple unpaired *t*-tests between HF and i-HF for each group and corrected for multiple comparisons using the Holm–Šídák method. *: *p* < 0.05, **: *p* < 0.01, ***: *p* < 0.001.

**Figure 4 cells-13-00849-f004:**
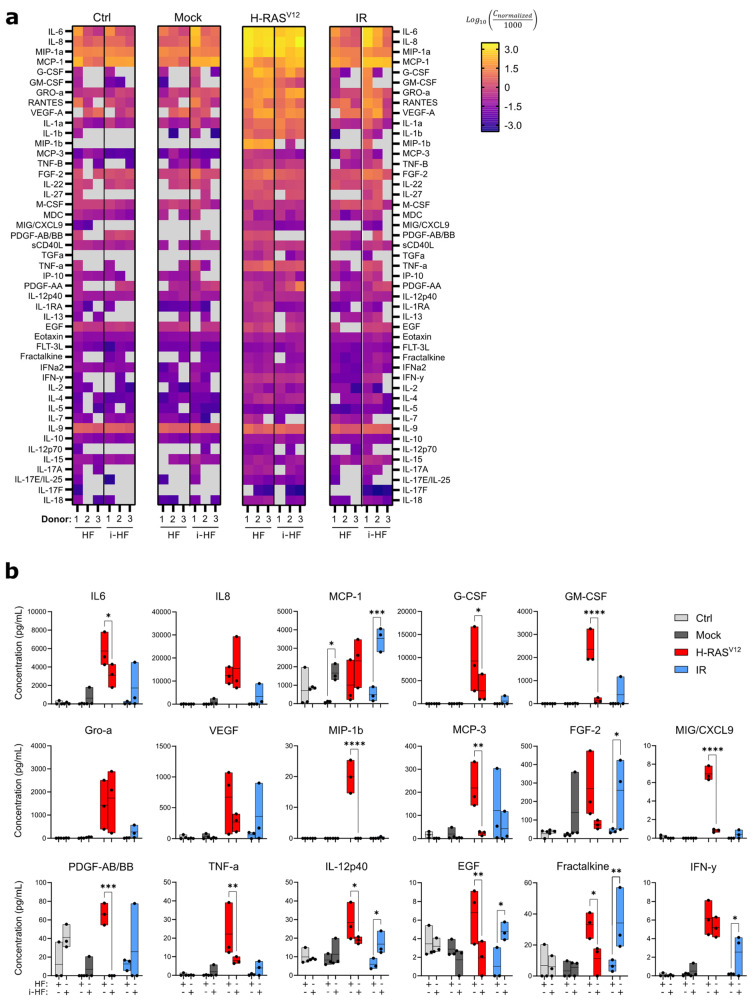
i-HF display different SASP profiles depending on the senescence inducer. (**a**) Heatmap of all cytokines detected from a multiplex assay performed using conditioned media collected from HF and i-HF (clone 1 from all 3 donors) 10 days after senescence induction with H-RAS^V12^ or exposure to 15 Gy IR. The results are represented as the log10 transformation of the normalized concentration after scaling the data on a common denominator. Gray squares indicate an undetected value. (**b**) Concentration (pg/mL) of the most abundant cytokines or those presenting notable differences between HF and i-HF. Each dot corresponds to one cell line. Undetected values were corrected to zero. Data analysis was performed using two-way ANOVA between HF and i-HF for each senescence inducer followed by Fisher’s LSD post hoc test. *: *p* < 0.05, **: *p* < 0.01, ***: *p* < 0.001, ****: *p* < 0.0001.

**Figure 5 cells-13-00849-f005:**
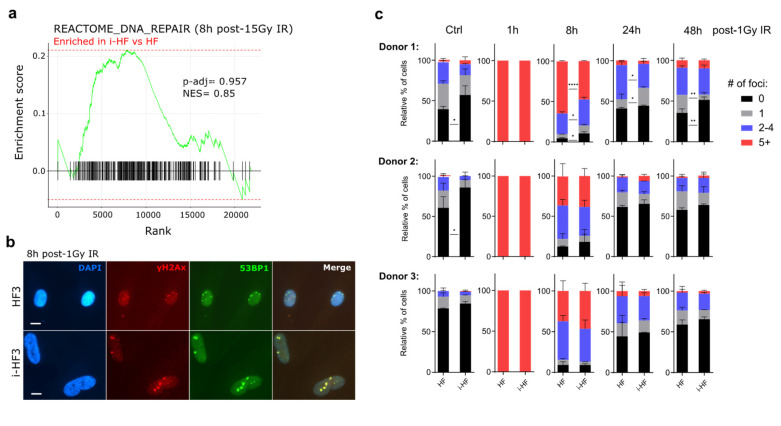
The DNA double-strand break repair capacity of i-HF is not compromised. (**a**) Gene set enrichment analysis (GSEA) and enrichment of gene sets associated with the Reactome_DNA_Repair was conducted using the pool of the three i-HF (clone 1) compared to HF donors at 8 h post exposure to 15 Gy IR. The adjusted *p*-value (*p*-adj) and normalized enrichment score (NES) are shown. (**b**) Representative images showing individual and merged signals from DAPI, γH2AX, and 53BP1 8 h post exposure to 1 Gy IR on HF from donor 3 are shown. Scale = 10 μM. (**c**) The relative number of cells presenting between 0 and 5 or more DNA damage foci at each timepoint post exposure to IR from all 3 donors is shown. Data analysis was performed using two-way ANOVA followed by Šídák’s multiple comparisons test. Shown is the mean ± SD of n = 2 independent experiments. (**d**) Graphical abstract representing the NHEJ reporter cassette inserted into HF that was reprogrammed and differentiated in i-HF. Shown is the digestion of I-SceI inverted recognition sequences after the nucleofection of cells with the indicated plasmids. (**e**) Shown is the proportion of cells expressing GFP and DsRed as determined via flow cytometry on HF4 and i-HF4 treated or not for 24 h with 10 μM of SAHA. (n = 3–6 independent experiments). (**f**) Representative DNA gel displaying unrepaired (635 bp) and repaired (429 bp—shown in red boxes) bands amplified from DNA collected 4 days after nucleofecting the I-SceI plasmid. (**g**) Shown is the proportion of DNA repair in I-SceI treated HF as determined by dividing the repaired band intensity by the sum of both bands (n = 3 independent experiments). Two-way ANOVA followed by Tukey’s multiple comparisons test was applied for the comparison of the GFP/DsRed ratio. A two-tailed unpaired *t*-test was applied between HF4 and i-HF4 levels for NHEJ repair from DNA band signals. Data are expressed as the mean ± SD. ns: not significant, *: *p* < 0.05; **: *p* < 0.01, ****: *p* < 0.0001.

**Figure 6 cells-13-00849-f006:**
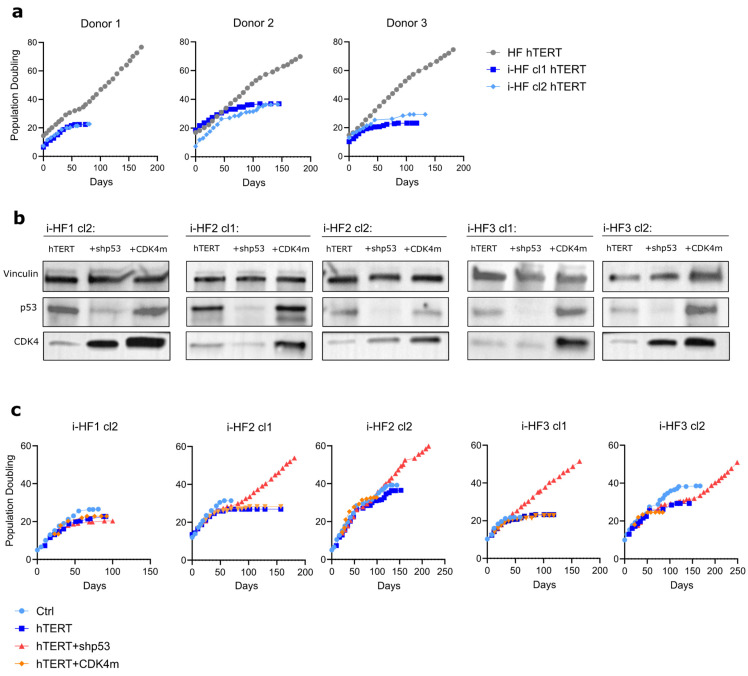
The immortalization of i-HF lines requires the knockdown of p53 expression. (**a**) Growth curve of HF and i-HF of the indicated cell lines as represented by PD counts initiated after transduction using hTERT lentiviral particles. (**b**) Protein expression of p53 and CDK4 from the indicated hTERT transduced cell lines subsequently transduced to express a shp53 or a mutant form of CDK4 unable to bind to p16 (CDK4m). Vinculin was used as a loading control. (**c**) Growth curve of the indicated cell lines as represented by PD counts.

## Data Availability

The raw sequencing data generated in this study have been deposited in the Gene Expression Omnibus (GEO) database under accession number GSE266663.

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
