# Peer review of "Induced Pluripotent Stem Cell-Derived Fibroblasts Efficiently Engage Senescence Pathways but Show Increased Sensitivity to Stress Inducers"

_cells, 2024, doi:10.3390/cells13100849_

Round 1
Reviewer 1 Report
Comments and Suggestions for Authors
The authors compared the parent fibroblasts and iPSCs-derived fibroblast phenotypes including their different responses to stimulants. It is interesting for the authors to observe decreased ability to get immortalized in iPSCs-derived fibroblasts in comparison to the skin fibroblasts. The following minor concerns need to be addressed to further improve the manuscript.
1. Figure 1c, what is the green color staining in immunofluorescence?
2. Figure 3d and 3f, extra lines were found.
3. Although the authors demonstrated the similar morphology and marker levels of skin fibroblasts and iPSCs-derived fibroblast, it is notable that iPSCs-derived fibroblasts do not demonstrate tissue-specificity, which means that the iPSCs-derived fibroblasts may not be most similar to skin fibroblasts, but resemble fibroblasts derived from other tissue types more. This may also be the reason for the differences between the skin fibroblasts and the iPSCs-derived fibroblasts in the manuscript. The authors may include this in the discussion part.
4. The authors may want to modify their description of malignant transformation of the iPSCs-derived fibroblasts. Although the authors used different stress to induce DNA damage, this is not equivalent to malignant transformation. Moreover, malignant transformation is reliant on rare cells that transform, but the authors only examined two clones of each iPSCs-derived fibroblast cell line, which is not sufficient to reach a conclusion that the iPSCs-derived fibroblasts do not demonstrate increased malignant transformation potential. This would be an overclaim.
Author Response
Review 1 comments:
- Figure 1c, what is the green color staining in immunofluorescence?
Response: There is no green color. The greenish shade is the result of the overlay of the blue and red color. Only few fibroblasts have very low expression of TRA1-60 and SSEA4.
- Figure 3d and 3f, extra lines were found.
Response: We apologize for this glitch in the uploaded figure. Not sure why this happened. The pdf figure was re-uploaded.
- Although the authors demonstrated the similar morphology and marker levels of skin fibroblasts and iPSCs-derived fibroblast, it is notable that iPSCs-derived fibroblasts do not demonstrate tissue-specificity, which means that the iPSCs-derived fibroblasts may not be most similar to skin fibroblasts, but resemble fibroblasts derived from other tissue types more. This may also be the reason for the differences between the skin fibroblasts and the iPSCs-derived fibroblasts in the manuscript. The authors may include this in the discussion part.
Response: We have added the following sentence in the discussion lines 546-547: Lack of terminal differentiation and maturation into skin-derived fibroblasts may explain such differences.
- The authors may want to modify their description of malignant transformation of the iPSCs-derived fibroblasts. Although the authors used different stress to induce DNA damage, this is not equivalent to malignant transformation. Moreover, malignant transformation is reliant on rare cells that transform, but the authors only examined two clones of each iPSCs-derived fibroblast cell line, which is not sufficient to reach a conclusion that the iPSCs-derived fibroblasts do not demonstrate increased malignant transformation potential. This would be an overclaim.
Response: We have removed the word malignant transformation and instead now use the term aberrant growth.
Reviewer 2 Report
Comments and Suggestions for Authors
A carefully designed and excellently conducted experiment comparing “parental” fibroblast with their induced-to-pluripotency- counterparts. The approach has the merit of focusing on discerning between possible pre existent/ induced genetic alteration that could act tumorigenic after transplantation, would the respective IPSCs be used as such
Several comments or rather suggestions are inserted below
Abstract
Very efficient in summarizing the presented work. Maybe adding two words regarding the method used for IPS induction would be of use.
Introduction
Is brief to the point clearly presenting the scientific rationale for conducting a study regarding the comparative expression of known marker of senescence and cell cycle inhibitors in induced and native skin fibroblasts under potential transformative conditions (ionizing radiation and overexpression of H-RAS oncogene).
Material and methods
Methods are clearly described with sufficient details
Why do the authors choose to use one donor with potential consistent epigenetic differences (fetal fibroblast) compared to the two others (adult male and female?) . Do they think rather a homogenous pool of donors would have helped to solidify the results? Would further comparison between fetal and adult fibroblast make sense?
Was there a particular reason why conditioned cell media for SASP release was collected 10 days after induction of senescence? Would a time-line expression of respective cytokines could have been more informative?
Elegant method to asses and compare DSB repair, adding to the complexity of interrogation and relevance of read outs and finding presented.
Results are systematically and well presented with relevant graphical support to enhance their understanding.
Regarding differences between transcriptomic profiling induced and native fibroblasts, do the authors consider induced one have a somehow less mature phenotype (given their increased proliferation rate and decreased ability to produce ECM and -probably overall lower propensity to express SASP?)
Interesting finding regarding donor dependent respective stress induced replicative senescence (as per p16 and p53 gene expression). Do the authors consider this finding could eventually generate a screening test for donor selection and/or prediction of possible IPSC based therapy outcomes?
Remarkably, one donor induced fibroblasts displayed reduced proliferation and beta Gal activity even after empty viral transduction, this is rather and important finding that maybe would require futher attention in terms of possible causality (epigenetic determinants?).
Interesting observation regarding the observation that adult but not fetal donor derived induced fibroblast appear to have fewer DNA damage immediately upon irradiation. This, again, would warrant further investigation regarding age related differences and similarity between donors.
Discussions
Donor related differences in transcriptomics profile in differentiated IPS is not a new finding (Reed, 2021, Sanchez-Freire 2014) and type of differentiation (homologous or non-homologous with parental cells) as well as protocol could further cause differences. Concentrating efforts in elaborating a screening panel to predict such differences in terms of senescence/maintenance of tumor suppression gene expression would be needed before translating IPS based products to clinic as authors suggest.
Overall, being aware of word limit imposed, this reviewer would still encourage authors to better reflect in the abstract (if not in the title as well) the complexity of methods employed to sustain their findings (transcriptomics and bioinformatics for example, maybe interrogation of DNA damage) that go beyond ordinary and consistently support author arguments for the findings.
Author Response
Review 2 comments:
- Very efficient in summarizing the presented work. Maybe adding two words regarding the method used for IPS induction would be of use.
Response: We have added the information in the abstract that cells were reprogrammed using the non-integrating Sendai virus method.
- Why do the authors choose to use one donor with potential consistent epigenetic differences (fetal fibroblast) compared to the two others (adult male and female?) . Do they think rather a homogenous pool of donors would have helped to solidify the results? Would further comparison between fetal and adult fibroblast make sense?
Response: This is an good question for which it is difficult to answer. Transcriptomic data revealed that adults and fetal cells are clustered together so we do not think there are major epigenetic differences. Moreover, in response to IR, donor 3 (fetal) and donor 2 (adult) behave more similarly than donor 1 (adult) vs donor 2 (adult). In fact, donor 1 turned out to be the most sensitive in activating the p16 pathway. In retrospective, while it would have been probably best to use 3 of a kind, the fetal sample still behave significantly more like adult HF than iHF.
- Was there a particular reason why conditioned cell media for SASP release was collected 10 days after induction of senescence? Would a time-line expression of respective cytokines could have been more informative?
Response: We choose to collect conditioned media at that time because it has been published before that the SASP is fully deployed at 7-10 days post induction of senescence (Coppe et al. Plos Bio 2008 – ref 28 in the manuscript). We have added this information in the method section 2.5.
- Regarding differences between transcriptomic profiling induced and native fibroblasts, do the authors consider induced one have a somehow less mature phenotype (given their increased proliferation rate and decreased ability to produce ECM and -probably overall lower propensity to express SASP?)
Response: Yes we agree with this interpretation of the data. This is in line with question #3 of reviewer 1. We added a sentence in the discussion indicating the probable lack of terminal differentiation/maturation of iHF lines.
- Interesting finding regarding donor dependent respective stress induced replicative senescence (as per p16 and p53 gene expression). Do the authors consider this finding could eventually generate a screening test for donor selection and/or prediction of possible IPSC based therapy outcomes?
Response: This would be possible only for non-autologous therapies. But yes in the context of using iPSC-derived cells differentiated from a universal donor iPSC line (lacking B2M genes for example). Such a screening test would be helpful to identify the best suited clone.
- Remarkably, one donor induced fibroblasts displayed reduced proliferation and beta Gal activity even after empty viral transduction, this is rather and important finding that maybe would require futher attention in terms of possible causality (epigenetic determinants?).
Response: Indeed iHF1 clone 1 showed very premature p16 expression which severely limited his growth/expansion potential. This clone was also prompt to activate this pathway following a stress such as viral transduction We do not think this is worth further attention as this basically shows the already well known high heterogeneity in iPSC clones (which we agree is likely epigenetic).
- Interesting observation regarding the observation that adult but not fetal donor derived induced fibroblast appear to have fewer DNA damage immediately upon irradiation. This, again, would warrant further investigation regarding age related differences and similarity between donors.
Response: Indeed, there is a trend towards fewer DNA damage in the fetal donor but we cannot make any conclusion with only one sample. We do not believe this warrants further investigation given the goal of our study was to compare native vs iPSC-derived cells, not to compare adult vs fetal clones.
- Overall, being aware of word limit imposed, this reviewer would still encourage authors to better reflect in the abstract (if not in the title as well) the complexity of methods employed to sustain their findings (transcriptomics and bioinformatics for example, maybe interrogation of DNA damage) that go beyond ordinary and consistently support author arguments for the findings.
Response: We added in the abstract that we used transcriptomics analysis as requested. That said, we also mention in the abstract that cells lines were transcriptionally different. Information about cellular senescence and DNA damage is also already mentioned.
Reviewer 3 Report
Comments and Suggestions for Authors
Induced pluripotent stem cells (iPSCs) are promising tool for regenerative medicine, but there are limitations of their clinical use, for example, risk of malignant transformation (caused by several factors including temporarily inactivation of the “guardian of genome”, p53, during reprogramming). The authors of this work investigated the differences between human fibroblasts (parental HF) and cells generated from them through the two-step process of generation of iPCSs via non-integrating Sendai viruses and differentiation of iPCSs to fibroblasts (iHF).
They found that parental HF from 3 donors and 3 corresponding i-HF are phenotypically similar, but significantly differ in transcriptomic signatures (indicating increased mitochondrial activity, ribosomal biogenesis, translation and “faster” cell cycle). In iHF, principal p53 effectors seemed to be unaffected by dedifferentiation/differentiation, and even increased in response to exposure to IR. The iHF demonstrated hyperactivated p53 and p21 effector functions. In response to DNA damage iHF entered senescence perhaps as efficiently as parental HF, but had different SASP profile depending on DNA damage inducer (IR or H-RASV12). DSB-repair was not compromised in iHF, NHEJ-repair was not impaired. Finally, iHF could not be immortalized with hTERT, unless p53 is knocked-down.
First and foremost, I find this manuscript very well-designed and written, its findings are novel and of great interest for whose working in the field, thus in my opinion it is suitable for the publication in its current state.
Please find my comments and questions below.
Line 86. “Single clones were manually picked and expanded on Matrigel (Corning, 354230) in E8Flex until stable with no/minimum of spontaneous differentiation”. - for how long (approximately) and how the spontaneous differentiation was assessed?
Line 143 “Senescence associated beta-galactosidase (SA-β-gal) expression” - just a note, SA-β-gal is not “an universal marker” of senescence, its presence does not necessarily indicate senescence. Thus, analysis of enescence-associated secretory phenotype (SASP) is a logical step.
Line 188. “under accession number XX” - please provide the accession number
Line 448. “Unexpectedly, we observed almost no GFP expression in i-HF4 compared to when using the parental HF4. These results lead us to hypothesize that i-HF4 may have acquired epigenetic modifications during the reprogramming process that led to the inactivation of the CMV promoter responsible for GFP expression” - What these modifications might be apart from histone deacetylation and what might be experimental approaches to test it?
Author Response
Reviewer 3 comments
- Line 86. “Single clones were manually picked and expanded on Matrigel (Corning, 354230) in E8Flex until stable with no/minimum of spontaneous differentiation”. - for how long (approximately) and how the spontaneous differentiation was assessed?
Response: Cells were expanded in E8Flex media on average for 4-6 passages until stable. Colonies were screened using an EVOS™ XL Core Imaging System and spontaneous differentiation eliminated by manual scrapping before passaging. We have added this information on line 90.
- Line 143 “Senescence associated beta-galactosidase (SA-β-gal) expression” - just a note, SA-β-gal is not “an universal marker” of senescence, its presence does not necessarily indicate senescence. Thus, analysis of enescence-associated secretory phenotype (SASP) is a logical step.
Response: Thanks for the precision. The text was modified as suggested
- Line 188. “under accession number XX” - please provide the accession number
Response: The number was not available at the time of submission. The number was added in this revised version.
- Line 448. “Unexpectedly, we observed almost no GFP expression in i-HF4 compared to when using the parental HF4. These results lead us to hypothesize that i-HF4 may have acquired epigenetic modifications during the reprogramming process that led to the inactivation of the CMV promoter responsible for GFP expression” - What these modifications might be apart from histone deacetylation and what might be experimental approaches to test it?
Response: This is an excellent question for which we do not have the answer. We used SAHA given it is a histone deacetylase inhibitor, but it also has broad epigenetic effect, including DNA demethylation. What are the possible other targets of SAHA we believe is outside the scope of this study.